# Overexpression of *PpSnRK1α* in Tomato Increased Autophagy Activity under Low Nutrient Stress

**DOI:** 10.3390/ijms23105464

**Published:** 2022-05-13

**Authors:** Jingjing Luo, Wenying Yu, Yuansong Xiao, Yafei Zhang, Futian Peng

**Affiliations:** State Key Laboratory of Crop Biology, College of Horticulture Science and Engineering, Shandong Agricultural University, Tai’an 271018, China; luo.jingj@163.com (J.L.); ying26231@163.com (W.Y.); ysxiao@sdau.edu.cn (Y.X.)

**Keywords:** SnRK1, autophagy, ABA, SnRK2, low nutrient tolerance

## Abstract

Plants suffer from a variety of environmental stresses during their growth and development. The evolutionarily conserved sucrose nonfermenting kinase 1-related protein kinase 1 (SnRK1) plays a central role in the regulation of energy homeostasis in response to stresses. In plant cells, autophagy is a degradation process occurring during development or under stress, such as nutrient starvation. In recent years, SnRK1 signaling has been reported to be an upstream activator of autophagy. However, these studies all focused on the regulatory effect of SnRK1 on TOR signaling and the autophagy-related gene 1 (ATG1) complex. In this study, overexpression of the gene encoding the *Prunus persica* SnRK1 α subunit (*PpSnRK1α*) in tomato improved the photosynthetic rates and enhanced the resistance to low nutrient stress (LNS). Overexpression of *PpSnRK1α* increased autophagy activity and upregulated the expression of seven autophagy-related genes (ATGs). The transcriptional levels of *SlSnRK2* family genes were altered significantly by *PpSnRK1α*, signifying that *PpSnRK1α* may be involved in the ABA signaling pathway. Further analysis showed that *PpSnRK1α* not only activated autophagy by inhibiting target of rapamycin (TOR) signaling but also enhanced ABA-induced autophagy. This indicates that *PpSnRK1α* regulates the photosynthetic rate and induces autophagy, and then responds to low nutrient stress.

## 1. Introduction

Different environmental challenges cause plants to suffer from various stresses and nutrient deprivation that disrupt metabolic and energy homeostasis, which are critical for plant development and growth. Plant sucrose nonfermenting kinase 1-related protein kinase 1 (SnRK1) is known as an energy and metabolism sensor that regulates the energy balance in response to nutritional and environmental stresses [1,2,3,4]. SnRK1 is a heterotrimeric complex composed of an α-catalytic subunit and β-and γ-regulatory subunits within plants [5,6]. Furthermore, evidence is accumulating that the α-subunit plays pivotal roles in regulating biotic and abiotic stress responses as well as plant growth and development [1,3,7,8,9].

In response to energy deficits, SnRK1 restores energy homeostasis by switching on ATP-producing catabolic pathways while also switching off ATP-consuming metabolic processes. *KIN10* (the gene encoding an α-catalytic subunit in *Arabidopsis thaliana*) overexpression in seedlings was shown to promote plant survival in glucose (Glc)-free liquid medium [7]. KIN10 promotes stomatal development, thereby influencing many plant responses to changing environmental conditions [10]. *PpSnRK1α* has been shown to be involved in the abscisic acid (ABA) signaling pathway to improve tomato salt tolerance [11].

In plant cells, various complex catabolic pathways are upregulated to balance the nutrient cycle and restore energy homeostasis under nutrient starvation stress, and autophagy is an important part of this process. Autophagy is an evolutionarily conserved self-degradation process that recycles cellular nutrients or breaks down damaged components for the optimization of plant growth, development and stress responses [12,13,14]. Nutrient deficiency induces the upregulation of the transcription levels of a large number of autophagy-related genes (ATGs) in plants, accompanied by the formation of autophagosomes. In Arabidopsis, *ATG3*, *ATG4a*, *ATG4b*, *ATG7* and *ATG8a-ATG8i* were shown to be induced by sugar starvation [15,16,17]. The ATG1 kinase complex, the main switch for autophagy initiation, is regulated by the upstream kinases target of rapamycin (TOR) and SnRK1 [18]. TOR is a phosphatidylinositol 3-kinase-related kinase that acts as a negative regulator of autophagy [19,20]. In general, TOR is activated and inhibits autophagy in nutrient-rich conditions, stimulating growth. Upon nutrient deprivation, TOR is inactivated, allowing for the activation of autophagy and reducing plant growth. Overexpression of *KIN10* in Arabidopsis leaf cells induces the expression of *ATG8s* [7], which play a key role in autophagosome formation and are very sensitive markers of autophagy [21]. KIN10 activates autophagy by inhibiting the TOR signaling pathway [22] or by directly phosphorylating ATG1 [23].

Plant hormones are important endogenous signaling compounds that regulate plant growth, development and response to stress. In Arabidopsis, autophagy is induced by the salicylic acid (SA) functional analog benzothiadiazole (BTH) [24,25]. Zeatin (ZT) can reduce the formation of autophagy in root epidermal cells [16], suggesting that phytohormones can directly regulate autophagy. Ethylene response factor 5 (ERF5) can directly bind to the ATG8d and ATG18h promoters to induce autophagy formation under drought stress [26]. Brassinosteroid (BR) signaling transcription factor BZR1 (Brassinazole resistance 1) can upregulate the expression of ATGs to induce autophagy in response to low nitrogen stress [27]. Thus, plant hormone signal transduction pathways are also involved in regulating autophagy. In Arabidopsis, ABA has been shown to activate the expression of *ATGs* and the formation of autophagosomes [28]. ABA-activated SnRK2 can phosphorylate RAPTOR under osmotic stress, thereby inhibiting TOR signaling to induce autophagy [29]. ABA promoted autophagy by enhancing autophagosome formation in glioblastoma cells [30].

Some reports indicate that SnRK1 links stress with ABA signaling [5,8]. SnRK1.1 is involved in ABA signaling pathways, resulting in modifications of essential enzyme activities and gene expression in response to sugar starvation and darkness [7,31]. In Arabidopsis, *SnRK1.1*-overexpressing lines exhibit an ABA-sensitive phenotype [31]. ABA-insensitive 1 (ABI1) and clade A type 2C protein phosphatases (PP2CAs), negative regulators of the ABA pathway, interact with the SnRK1 α subunit to repress SnRK1 signaling [4]. Apple MdSnRK1.1 (α-catalytic subunit) interacts with MdCAIP1 to regulate ABA sensitivity [32]. However, the role of SnRK1 in ABA-induced autophagy is largely unknown.

Our previous study showed that overexpression of *PpSnRK1α* in tomato could promote fruit ripening and enhance salt tolerance. In this study, it was found that overexpressing *PpSnRK1α* in tomato increased the photosynthetic rate and autophagy activity under low nutrient tolerance. Further research found that *PpSnRK1α* not only enhanced autophagy through TOR signaling but also enhanced ABA-induced autophagy. Our research places *PpSnRK1α* as an important regulator controlling the balance between growth and stress responses via the direct regulation of autophagy.

## 2. Results

### 2.1. Effects of PpSnRK1α Overexpression on Tomato Photosynthetic Rate and Plant Growth under LNS

Plant SnRK1 protein kinase is a central energy and metabolism sensor that regulates the energy balance in response to biotic and abiotic stress. *PpSnRK1α*-overexpressing tomato (OE) and WT tomato were used to study the effects of SnRK1 on the growth of tomato plants under nutrient stress. In the control group, the SnRK1 activity in the leaves of OE plants was 30.23% higher than that of WT plants. Compared with tomato under normal nutrient levels (control), OE tomato under LNS had higher SnRK1 activity and was 39.46% higher than that of the WT (Figure 1A). Further, we analyzed the effects of *PpSnRK1α* on leaf photosynthesis. Compared with that of WT plants, the net photosynthetic rate of functional leaves of OE tomato increased by 9.79% on average under normal nutrient conditions (control) and 18.98% under LNS conditions (Figure 1B). As shown in Figure 1C, the relative chlorophyll content in leaves of OE plants was significantly higher than that of WT plants, which was 22.04% higher than that of WT plants after 12 days under LNS conditions. Compared with WT tomato plants, OE tomato plants had higher plant height and greater root and leaf weights (Table 1). These results indicated that *PpSnRK1α* could increase leaf photosynthesis and promote plant growth.

### 2.2. Effects of PpSnRK1α Overexpression on the Leaf Antioxidant Enzyme Activities

In the LNS group, the leaves of tomato plants were yellow and obviously smaller than those of plants in the control group, but the leaves of OE plants were obviously greener than those of WT plants, suggesting that *PpSnRK1α* may affect the process of leaf senescence under LNS (Figure 2A). Furthermore, we monitored changes in antioxidant enzyme activities and MDA content in both WT and OE plants before and after the LNS treatment. Compared with that in control leaves, the MDA contents in OE and WT leaves increased at 6 and 12 days after LNS treatment, but the MDA content in OE tomato leaves was lower than that in WT tomato leaves, indicating that *PpSnRK1α* overexpression leads to a decrease in the degree of cell membrane peroxidation in response to LNS (Figure 2B). The activities of antioxidant enzymes increased in response to stress at the initial stage of LNS and decreased as the stress time increased. Under both the control and LNS conditions, the activities of SOD, POD and CAT in the leaves of OE plants were significantly higher than those in the leaves of WT plants (Figure 2C–E). These results indicated that *PpSnRK1α* may play a role in antioxidant systems to protect plants under LNS.

### 2.3. Overexpression of PpSnRK1α Enhances Autophagy Activity under LNS

Autophagy is a highly conserved self-degradation mechanism in eukaryotes. Under nutritional stress conditions, autophagy is largely induced, and plants can maintain cell homeostasis through autophagy. To investigate the effect of *PpSnRK1α* on autophagy under LNS, we measured the expression of ATGs and autophagy activity in OE and WT tomatoes. We found that 7 *SlATGs* were induced, and 3 *SlATGs* were inhibited by LNS relative to normal conditions (LNS vs. control). Under LNS conditions, the expression levels of *SlATG1*, *SlATG2*, *SlATG4*, *SlATG5*, *SlATG6*, *SlATG9* and *SlATG11* in OE plants were higher than those in WT plants, while the expression levels of *SlATG3a*, *SlATG8* and *SlATG12* were lower than those in WT plants (Figure 3A). These results suggest that autophagy in tomato leaves could be induced by LNS and that *PpSnRK1α* may mediate the formation of autophagy under LNS.

Furthermore, we used the fluorescent dye MDC to detect autophagic activity in WT and OE plants under control and LNS conditions (Figure 3B). Under the control condition, only a few autophagosomes were detected in WT and OE tomato leaves. After 7 days of LNS treatment, the number of autophagosomes in WT and OE leaves was significantly increased compared with that in the control group, and the number of autophagosomes in OE tomato leaves was 1.47 times that in WT leaves (Figure 3C), confirming that *PpSnRK1α* plays a positive role in low nutrient-induced autophagy. 

### 2.4. Overexpression of PpSnRK1α Affects the ABA Content and Transcription Level of SlSnRK2s under LNS

The ATG1 kinase complex, as the main switch of autophagy initiation, is regulated by the upstream kinases TOR and SnRK1 [18]. It was found that the overexpression of PpSnRK1α could improve the resistance of tomato to LNS, accompanied by an increase in autophagy activity. In *Arabidopsis thaliana*, downregulation of TOR activates autophagy under nutritional stress. Thus, we further explored the role of TOR in PpSnRK1α-induced autophagy. The TOR activator Glc was sprayed on the tomato leaves, and the autophagy activities in WT, WT + Glc, OE and OE + Glc leaves were measured by MDC staining. As shown in Figure 4A,B, the autophagy activity in WT and OE leaves significantly decreased after treatment with the TOR activator Glc. Under normal nutrient conditions, no MDC-labeled autophagosomes were detected in WT + Glc and OE + Glc leaves. Under LNS conditions, the number of MDC-stained autophagosomes in WT + Glc leaves was 36.4% lower than that in WT leaves, while the number of autophagosomes in OE + Glc leaves was 45.8% lower than that in OE leaves (Figure 4). Overall, these data suggested that the activation of *PpSnRK1α* on autophagy activity depended on TOR signaling and that *PpSnRK1α* was involved in the regulation of other signal-induced autophagy.

ABA, an essential plant hormone, plays a crucial role in response to abiotic stress, and the negative regulators of ABA signaling, ABI1 and PP2CA, directly inhibit the activity of SnRK1. Thus, we speculated that ABA signaling may induce autophagy and that SnRK1 may be involved in the ABA-induced autophagy process. SnRK2, a subfamily of kinases related to the SnRK1 subfamily, is an important component of the ABA signaling pathway. Therefore, the ABA content and the expression levels of seven *SlSnRK2s* were detected in both OE and WT plants (Figure 5). There was no significant difference in ABA content between WT and OE tomato leaves, but the ABA content was significantly increased under LNS conditions, which was 3.46 times that under control conditions (Figure 5A). This suggested that *PpSnRK1α* was downstream of ABA signaling. In OE plants, the transcript levels of *SlSnRK2.3*, *SlSnRK2.4*, *SlSnRK2.6* and *SlSnRK2.7* were increased under both control and LNS conditions, while the transcript level of *SlSnRK2.2* was inhibited. Compared with the control, the expression level of *SlSnRK2.1* was not significantly changed in the WT under LNS conditions but was increased significantly in the OE lines. Under LNS conditions, the expression levels of *SlSnRK2.3*, *SlSnRK2.4*, *SlSnRK2.6* and *SlSnRK2.7* were upregulated, while those of *SlSnRK2.2* and *SlSnRK2.5* were downregulated (Figure 5B–H). These results suggest that the regulation of *SlSnRK2* transcription by *PpSnRK1α* is complex.

### 2.5. Overexpression of PpSnRK1α Enhances ABA-Induced Autophagy

To investigate whether ABA can induce autophagy in tomato leaves, we used MDC to detect autophagy activity after treatment with exogenous ABA. As shown in Figure 6, autophagy activity was significantly increased after ABA treatment, especially treatment with 30 µM ABA (Figure 6A,B). To gain further insight into the role of ABA in activating autophagy, we analyzed the transcript levels of 12 tomato *ATGs* after treatment with ABA (Figure 6C). The transcript levels of *SlATG2*, *SlATG5*, *SlATG6*, *SlATG7*, *SlATG8c*, *SlATG10*, *SlATG12* and *SlATG13* were significantly upregulated, and only two *ATGs*, *SlATG4* and *SlATG9*, were downregulated. These results, taken together, confirm that exogenous ABA can induce the formation of autophagosomes in tomato leaves.

ABA can control SnRK1 activity via the interaction of SnRK1 with PP2C to regulate its phosphorylation [4]. In this study, the expression of *PpSnRK1α* in OE leaves was also detected at different time points after 0 µM and 30 µM ABA treatments. The expression of *PpSnRK1α* increased after ABA treatment and reached the highest level at 12 h, which was 3.7 times that before treatment (Figure 7A), indicating that ABA could also regulate PpSnRK1α transcription. Therefore, we speculated that *PpSnRK1α* might be involved in ABA-induced autophagy. To determine whether *PpSnRK1α* is involved in the regulation of ABA-induced autophagy, WT and OE tomato leaves were treated with 30 μM ABA, and autophagy activity was detected by MDC staining. As shown in Figure 7B,C, the number of autophagosomes in WT and OE leaves increased significantly after ABA treatment. Under LNS conditions, the number of autophagosomes in WT and OE leaves increased by 32.4% and 41.6%, respectively, suggesting that the overexpression of *PpSnRK1α* enhanced the ABA-dependent activation of autophagy under LNS.

## 3. Discussion

The role of SnRK1 has been researched mostly in *A. thaliana* and crop species, while the role of SnRK1 in horticultural crops has rarely been reported. In this study, we overexpressed peach *PpSnRK1α* in tomato to study the mechanism of *PpSnRK1α* in response to LNS. In Arabidopsis, SnRK1 can be activated by sugar starvation, energy deficiency, dark culture and oxygen deficiency. KIN10/11 can initiate multiple transcription cascades in response to sugar or energy depletion in darkness and under stress conditions. The role of KIN10 in the starvation response was most evident in Glc-free liquid medium [7]. SnRK1 activity was shown to be higher in seedlings grown under low nutrient conditions than in seedlings grown under optimal nutrient conditions [33]. Consistent with these findings, SnRK1 activity was higher in OE tomatoes grown under LNS conditions than in OE tomatoes grown under control conditions, and SnRK1 activity in OE lines increased more evidently.

Mounting evidence has shown that SnRK1 plays a key role in carbohydrate metabolism and maintains energy homeostasis [34,35]. SnRK1 is also involved in starch biosynthesis [36,37]. In our previous study, transgenic tomato plants overexpressing heterologous *MhSnRK1* had greater soluble sugar and starch contents and greater photosynthesis rates than WT plants [37]. In the photosynthesis pathway, seven genes were upregulated in the *MhSnRK1*-overexpressing lines [38]. SnRK1 also plays a role in nitrogen metabolism [39]. In this study, the photosynthetic rate and the activity of antioxidant enzymes in mature leaves of OE were increased significantly compared with WT. After 12 days under LNS, the transgenic lines showed higher root biomass and better root activity. These results suggest that *PpSnRK1α* overexpression leads to enhanced resistance to low nutrient stress in tomatoes.

Autophagy is a highly conserved self-degradation mechanism in eukaryotes, and the cascade regulation of phosphorylation is the main switch to activate autophagy [40]. Autophagy usually begins with the perception of developmental and nutritional signals by the ATG1 complex, and ATG1 is regulated by the upstream kinases TOR and SnRK1 [18]. In Arabidopsis, autophagy is induced during senescence and nutrient deficiency [41,42] and stresses [43,44] by inhibiting TOR pathways. Overexpression of *TOR* inhibited autophagy activation by nutrient starvation and salt and osmotic stress [45]. SnRK1 can induce autophagy both via TOR-dependent and TOR-independent pathways in Arabidopsis [22]. In Arabidopsis, SnRK1 phosphorylation of RAPTOR represses TOR complex activity to activate autophagy in plants [46,47]. Overexpression of KIN10 activates the formation of autophagosomes by increasing the level of ATG1 phosphorylation, and KIN10 can interact with ATG1a and ATG13a in vitro, suggesting that SnRK1 can also regulate autophagy through the ATG1 complex. Transgenic Arabidopsis lines overexpressing *KIN10* increased tolerance to nutrient starvation [23]. Tomato SnRK1.1 mediates the formation of ATG6-dependent autophagy and alleviates tomato LN stress [48]. In this study, *PpSnRK1α*-overexpressing lines exhibited higher autophagy activity than WT plants under LNS and showed stronger resistance to LNS. Moreover, overexpression of *PpSnRK1α* upregulated the expression levels of seven ATGs. Given that the TOR complex is also a major regulatory factor in nutrient sensing, we added the TOR activator Glc to verify whether SnRK1-mediated autophagy under LNS is related to the TOR pathway. After TOR was activated by Glc, autophagy activity was decreased in OE leaves; however, some autophagosomes were still observed, indicating that *PpSnRK1α* can activate autophagy activity through TOR-dependent and TOR-independent pathways. Although *PpSnRK1α* plays a positive role in autophagy regulation, whether autophagy induction is the cause of enhanced low nutrient tolerance requires further investigation, such as overexpressing *PpSnRK1α* in an autophagy-deficient context (mutant plants, treatment with inhibitors) to assess the role of autophagy in this phenotype.

Plants suffer from a variety of environmental stresses, during which ABA signaling and autophagy processes are activated to enhance stress tolerance. Therefore, we hypothesized that the ABA signaling pathway is involved in the regulation of autophagy under abiotic stress. In Arabidopsis, ABA activated the formation of autophagosomes and the expression of ATGs [28]. A recent study showed ABA-induced autophagy in glioblastoma cells [30]. Similarly, in this study, autophagosomes stained with MDC were observed in WT tomato leaves after treatment with different concentrations of ABA. Moreover, the expression levels of seven ATGs were significantly increased, and those of two ATGs were significantly decreased after 30 μM ABA treatment. These results confirmed that ABA activates autophagy.

ABA can respond to biotic and abiotic stresses [49], and ABI1 and PP2CA directly regulate SnRK1.1 activity [4]. Therefore, we hypothesized that SnRK1 may be involved in ABA-induced autophagy. The expression of *PpSnRK1α* was significantly increased after 30 μM ABA treatment, and peach seedlings overexpressing *PpSnRK1α* were sensitive to ABA [50]. Similar results were also found in apples, in which MdSnRK1.1 inhibited C2-domainABA insensitive protein1 (MdCAIP1)-mediated ABA sensitivity [32]. In this study, we sprayed 30 μM ABA on the leaves of WT and OE tomatoes growing under normal and LNS conditions and found that the autophagy activity in WT and OE leaves increased significantly after ABA treatment. More importantly, the autophagy activity increased more in OE leaves than in WT leaves after ABA treatment. These results suggested that SnRK1 is a positive regulator of ABA-activated autophagy.

SnRK2s play an essential role in response to biotic and abiotic stress [51]. In Arabidopsis, SnRK2.2, 2.3 and 2.6 play critical roles in ABA signaling [52,53]. Similar to ATGs, we found that the expression levels of *SlSnRK2s* were also regulated by *PpSnRK1α*. Under LNS conditions, the ABA content in tomato leaves increased significantly, and overexpression of *PpSnRK1α* significantly upregulated the expression levels of *SlSnRK2.1*, *SlSnRK2.3*, *SlSnRK2.4*, *SlSnRK2.6* and *SlSnRK2.7*. In the absence of ABA, ABA-responsive SnRK2s are usually inhibited by PP2Cs. In the presence of ABA, SnRK2s are released and activated to regulate the expression of ABA-responsive genes [54,55,56]. Within the *SlSnRK2* gene family, *SlSnRK2.3* and *SlSnRK2.4* belong to SnRK2 subclass III and are essential for transducing ABA signals [57,58]. In conclusion, PpSnRK1α and SlSnRK2s crosstalk can regulate the response of tomato to LNS, and the regulation of SlSnRK2s by *PpSnRK1α* may be related to the ABA signaling pathway. However, whether there is a direct or indirect interaction between *PpSnRK1α* and SnRK2 to regulate the plant response to LNS needs further study.

## 4. Summary

Overexpressing *Prunus persica PpSnRK1α* in tomato increased photosynthetic rate and induced autophagy under low nutrient tolerance. Moreover, *PpSnRK1α* not only enhanced autophagy through TOR signaling but also played a positive role in ABA-induced autophagy. However, the mechanism by which SnRK1 responds to LNS is complex, and the mechanism by which SnRK1 is involved in ABA-induced autophagy and whether SnRK1 directly regulates *SlSnRK2* expression need further investigation.

## 5. Materials and Methods

### 5.1. Plant Materials and Treatments

In this study, wild-type (WT) tomato (*Solanum lycopersicum* ‘Sy12f’) and homozygous T2 lines overexpressing *PpSnRK1α* (ppa004347 m) (OE) [59] were germinated and grown in a plant growth chamber filled with a mixture of peat and vermiculite (1:1, *v*/*v*) at 23 °C for 3 weeks.

For the low nutrient stress (LNS) study, 1/2 Hoagland nutrient solution was used as the normal nutrient level (control), and 1/20 Hoagland nutrient solution was used as the LNS level. The 3-week-old plants were moved to a 10 L hydroponic system containing 8 L of 1/2 Hoagland nutrient solution, and the treatments were performed after 1 week of growth. The hydroponic system was ventilated 24 h a day, and the nutrient solution was replaced every 3 days.

For treatment with Glc, during the LNS treatment, the tomato plants were sprayed with 15 mM Glc solution or ddH_2_O every 2 days. Autophagy activity was detected on the 7th day of treatment.

For treatment with ABA, 6-week-old WT leaves were sprayed with 0, 10, 20, 30 or 50 µM ABA. Twelve hours after ABA treatment, the upper first fully expanded leaves were excised to detect autophagy activity, or they were sampled and frozen quickly in liquid N and stored at −80 °C before being used for gene expression.

For treatment with ABA under control and LNS conditions, the tomato plants were sprayed with 0 µM and 30 µM ABA solutions every 2 days. Samples were collected at different time points after ABA treatment to detect the expression level of *PpSnRK1α*. Autophagy activity was detected on the 7th day of treatment.

### 5.2. Determination of SnRK1 Activity

Approximately 1.0 g of frozen, fine tissue powder was homogenized in 1 mL of cold buffer solution that consisted of 100 mM HEPES (pH 8), 25 mM NaF, 2 mM sodium pyrophosphate, 0.5 mM EDTA, 0.5 mM EGTA, 5 mM DTT, 1 mM anisole, 1 mM phenylmethylsulfonyl fluoride, 1 mM protease inhibitor cocktail (Sigma P9599), phosphatase inhibitors (PhosStop; Roche) and insoluble polyvinylpyrrolidone (PVP). The tissue homogenate was transferred to a cold microfuge tube and clarified by centrifugation at 12,000× *g* for 5 min at 4 °C. The supernatant (750 µL) was desalted on a 2.5 mL pre-equilibrated centrifuge column (Sephadex G-25 medium columns; GE Healthcare) and maintained at 4 °C for the SnRK1 activity assay. SnRK1 activity was determined with a Universal Kinase Activity Kit (R&D Systems, Minneapolis, MN, USA, EA004) by using AMARA polypeptide as the substrate [59]. The procedure was carried out according to the manufacturer’s instructions. The optical density at 620 nm (OD620) was measured after the enzymatic reaction was over, and the relative SnRK1 activity was calculated based on OD620.5.3. Determination of antioxidant enzyme activities and the MDA content.

Proteins were extracted from approximately 0.5 g of sample with 4 mL of cold 50 mM phosphate buffer solution (pH = 7.8) [60]. Superoxide dismutase (SOD) activity was determined by the nitrogen blue tetrazole (NBT) method and calculated according to the method of Wang et al. [61]; catalase (CAT) activity was determined using the ultraviolet absorption method as previously described [62], and peroxidase (POD) activity was determined using the guaiacol method [63]. SOD, POD and CAT activities are expressed as U·min^−1^ g^−1^ (fresh weight, FW). The malondialdehyde (MDA) content was determined by the thiobarbituric acid (TBA) method [64].

### 5.3. Measurement of Photosynthetic Rate and Chlorophyll Content

The daily variation in the photosynthetic rate in tomato leaves was measured on clear days with a CIRAS-3 portable photosynthetic measurement system (Hansha Scientific Instruments Limited, Tai’an, China). The chlorophyll content was measured using a SPAD-502 chlorophyll meter (Tuopu Yunnong Technology). Five mature leaves were measured following the instrument instructions, and the mean of these measurements was used to represent the photosynthetic rate and chlorophyll content of that individual plant.

### 5.4. RT-qPCR

Total RNA was extracted from evenly mixed samples (1 g, the sample is a mixture of each separately treated sample) using a Fast Pure Plant Total RNA Isolation Kit (RC401, RC401, Vazyme, Nanjing, China) and reverse transcribed to generate the first-strand cDNA using HiScript III RT SuperMix for qPCR (+gDNA wiper) (R323-01, Vazyme, Nanjing, China) according to the manufacturer’s instructions. Quantitative real-time PCR (RT-qPCR) consisted of three biological and three technical replicates and was carried out using the Bole CFX96 system (Bio-Rad, Hercules, CA, USA) and Ultra SYBR mixture (CW2601M, CWBIO, Beijing, China,). The calculation method for RT–qPCR was 2^−ΔΔCt^ with *FaACTIN* as the internal control. The specific primers used are listed in Appendix A.

### 5.5. MDC Staining

Tomato leaves were stained with monodansylcadaverine (MDC) (Sigma–Aldrich, 30432) to determine autophagy activity, as described by Wang et al. [65]. Tomato leaves approximately 2 mm × 4 mm in size were excised, immediately vacuum infiltrated with 100 μM MDC (Sigma–Aldrich 30432) for 30 min and then washed twice with phosphate-buffered saline. MDC-incorporated structures were observed with a confocal laser scanning microscope.

### 5.6. ABA Extraction and Determination

Collected samples were homogenized using liquid nitrogen, and 0.3 g of plant material was extracted in a 10 mL centrifuge tube for 24 h using 80% methyl alcohol. The homogenate was centrifuged at 8000 rpm at 4 °C for 10 min, and the supernatant was transferred to a distillation bottle and concentrated to dryness under vacuum. The residue was dissolved in phosphoric acid buffer (pH = 8) and chloroform and transferred to a 10 mL centrifuge tube. The extract was shaken for 30 min and then incubated for at least 30 min. After stratification, the lower layer of trichloromethane was pipetted to remove the pigment. The upper phase was collected, and 150 mg of PVP was added and centrifuged at 8000 rpm at 4 °C for 10 min. The supernatant was transferred to a centrifuge tube and extracted with 5 mL of ethyl acetate. After stratification, the upper phase was collected and concentrated to dryness under vacuum. The residue was dissolved in 1 mL of 0.04% (*v*/*v*) acetic acid/methanol (55:45, *v*/*v*) and filtered through a 0.45 µm filter. ABA quantification was performed on three independent samples by high-performance liquid chromatography (HPLC) (10 μL of sample was injected). ABA quantification was performed on three independent samples by high-performance liquid chromatography (Agilent 1100). The extracts were filtered through 0.22-µm syringe filters and directly injected through a 10 µL fixed loop into a C18 guard column (4.6 mm × 250 mm, 5 µm, Kromasil, Sweden). The mobile phase was the mixture of 0.1% acetic acid: methanol (*v*:*v* = 1:1). The ABA contents were quantified from the areas of their peaks at 254 nm using external standard calibration curves.

## Figures and Tables

**Figure 1 ijms-23-05464-f001:**
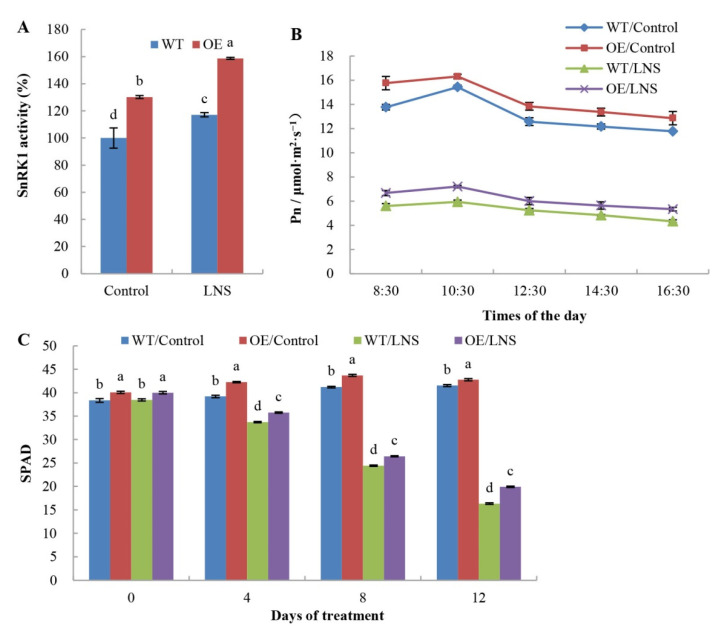
Effects of *PpSnRK1α* overexpression on the photosynthetic capacity of tomatoes under LNS. (**A**) SnRK1 activity in the leaves of *PpSnRK1α*-OE and WT tomatoes 7 days after LNS treatment. (**B**) Photosynthetic rates in mature leaves of the WT and OE line tomatoes 7 days after LNS treatment. (**C**) The dynamics of leaf SPAD readings of the WT and OE tomato lines. The data are shown as the means ± SDs of three independent biological replicates. Different lowercase letters in the same days of treatment indicate significant differences according to Duncan’s multiple range test (*p* < 0.05). Control: 1/2 Hoagland nutrient solution (normal nutrient level); LNS: 1/20 Hoagland nutrient solution (low nutrient level).

**Figure 2 ijms-23-05464-f002:**
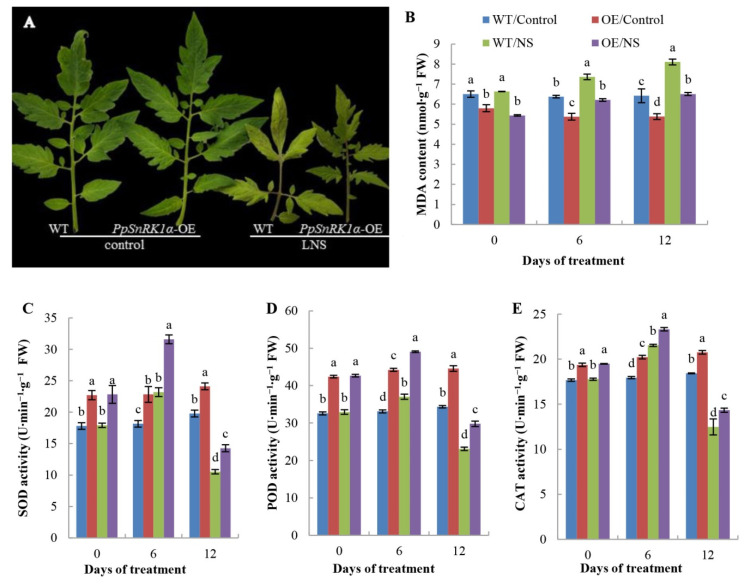
Response of wild-type (WT) and *PpSnRK1α*-overexpressing (OE) plants to low nutrient stress (LNS). (**A**) Phenotypes of the third functional leaf from the top of tomato plants after 12 days of treatment. (**B**–**E**) The MDA content (**B**), SOD (**C**), POD (**D**) and CAT (**E**) activities in the leaves of WT and OE lines. Values are the mean ± SD of triplicate samples. Different lowercase letters on the same days of treatment indicate significant differences according to Duncan’s multiple range test (*p* < 0.05).

**Figure 3 ijms-23-05464-f003:**
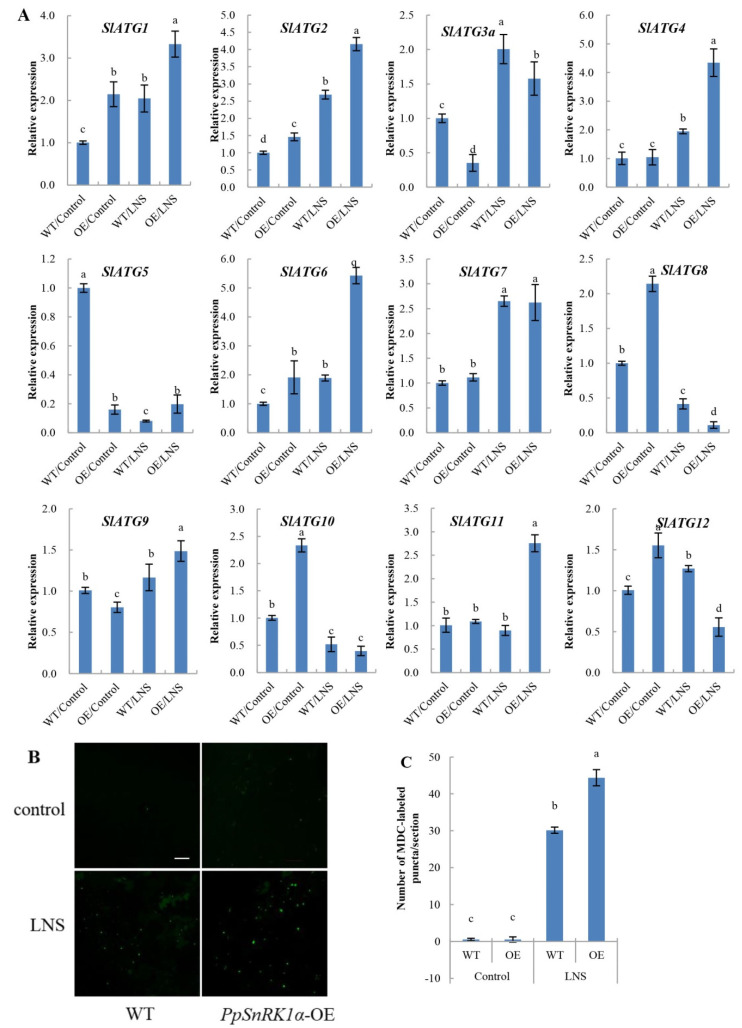
Effects of *PpSnRK1α* overexpression on autophagy activity in WT and OE leaves under 7 days of LNS. (**A**) Changes in the transcription levels of tomato ATGs in the WT and OE lines. (**B**) Autophagosomes by MDC staining in leaves of WT and OE plants. Autophagosomes by MDC staining are shown as green signals. Bars: 50 μm. (**C**) The number of autophagosomes by MDC staining per image in (**B**) was quantified to calculate the autophagic activity. Different lowercase letters indicate significant differences according to Duncan’s multiple range test (*p* < 0.05).

**Figure 4 ijms-23-05464-f004:**
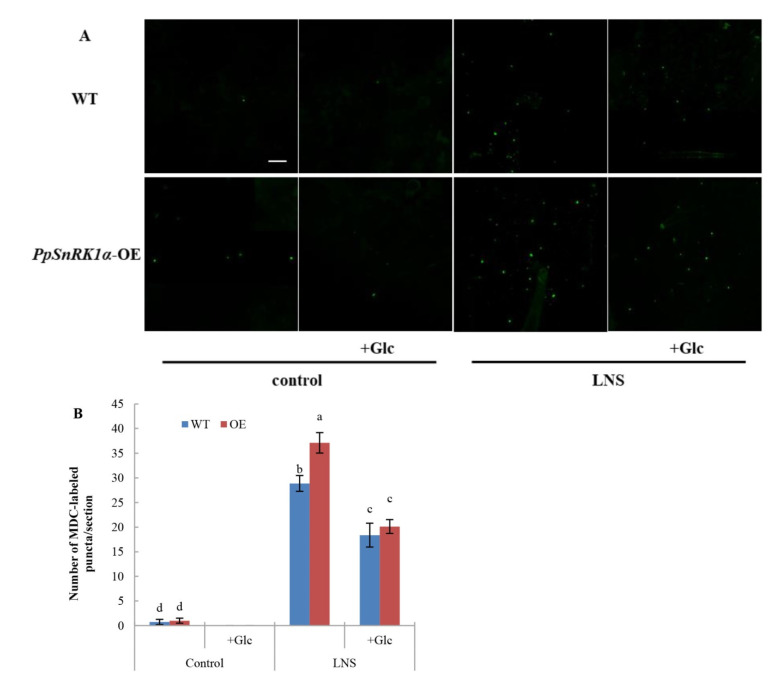
Activation of TOR by Glc inhibits autophagy. (**A**) Autophagosomes by MDC staining in leaves of WT and OE plants. Autophagosomes by MDC staining are shown as green signals. Bars: 50 μm. (**B**) The number of autophagosomes by MDC staining per image in (**A**) was quantified to calculate the autophagic activity. Different lowercase letters indicate significant differences according to Duncan’s multiple range test (*p* < 0.05).

**Figure 5 ijms-23-05464-f005:**
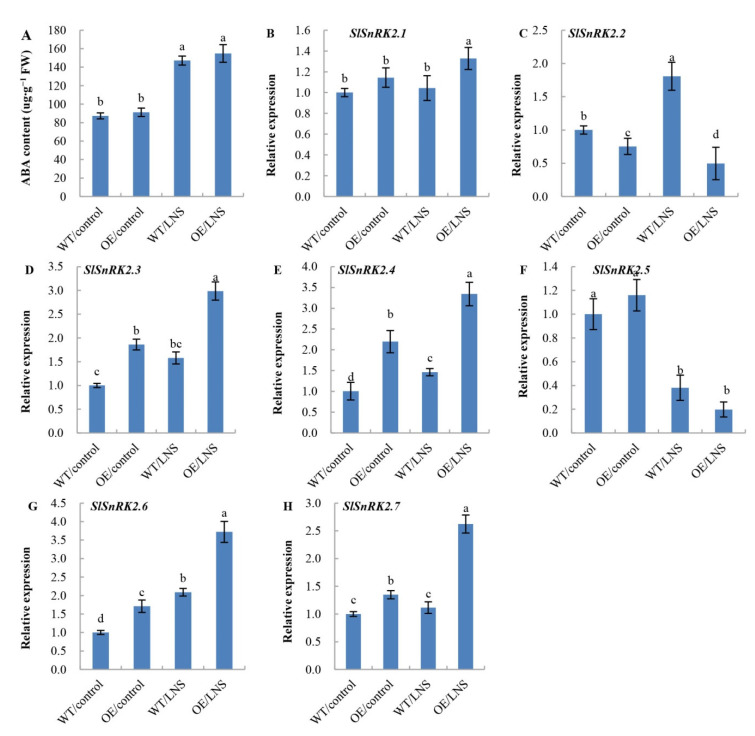
Effects of *PpSnRK1α* overexpression on ABA content (**A**) and the transcription level of SlSnRK2s (**B**–**H**). Data are expressed as the mean ± SD of biological triplicates. Different lowercase letters indicate significant differences according to Duncan’s multiple range test (*p* < 0.05).

**Figure 6 ijms-23-05464-f006:**
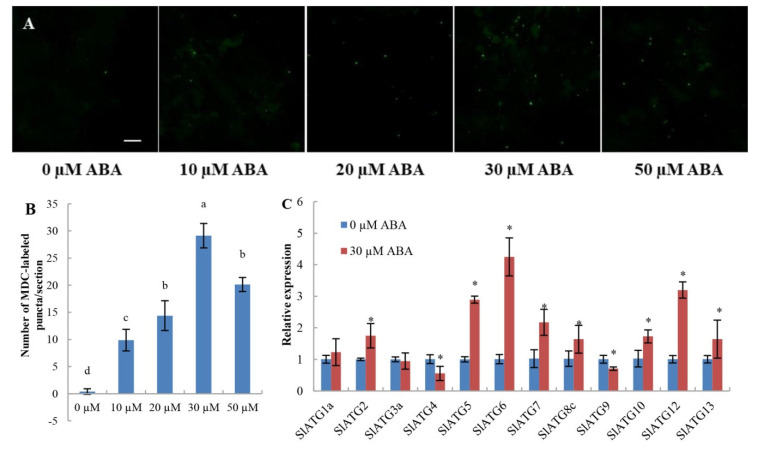
Effects of ABA on the induction of autophagy in tomato leaves. (**A**) MDC-stained autophagosomes in the leaves of WT tomatoes. Six-week-old plants were treated with 0, 10, 20, 30 or 50 μM ABA, and the leaves were stained with MDC and visualized at 12 h by confocal microscopy. Autophagosomes by MDC staining are shown as green signals. Bars: 50 μm. (**B**) The number of autophagosomes by MDC staining per image in (**A**) was quantified to calculate the autophagy activity. Data are expressed as the mean ± SD of seven biological replicates. Different lowercase letters indicate significant differences according to Duncan’s multiple range test (*p* < 0.05). (**C**) Relative expression of *SlATGs* in WT tomato leaves after 0 μM and 30 μM ABA treatment. Data are expressed as the mean ± SD of biological triplicates. An asterisk (*) on top of the error bar indicates a significant difference between different treatments at *p* < 0.05.

**Figure 7 ijms-23-05464-f007:**
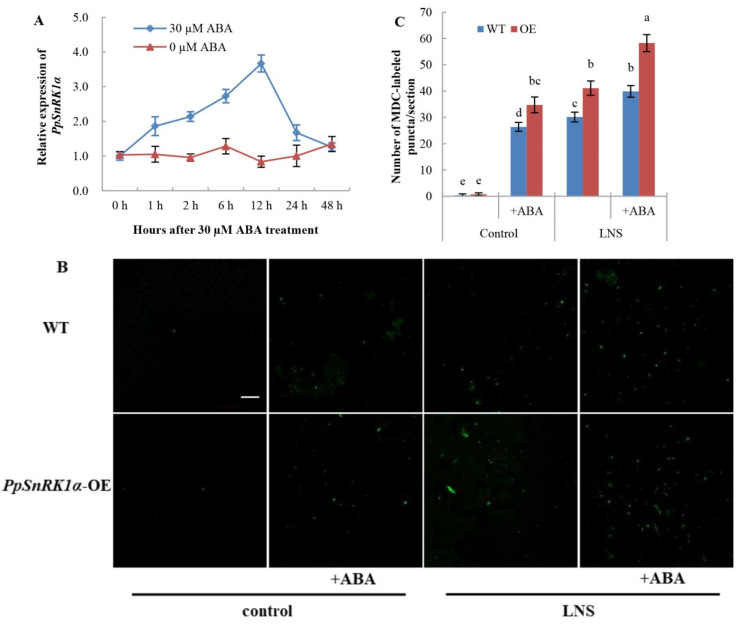
Effects of ABA on autophagy induction in tomato leaves. (**A**) Relative expression of *PpSnRK1α* in OE leaves after 0 μM and 30 μM ABA treatment. (**B**) MDC-stained autophagosomes in the WT and OE leaves under LNS conditions. Autophagosomes by MDC staining are shown as green signals. Bars: 50 μm. (**C**) The number of autophagosomes by MDC staining per image in (**B**) was quantified to calculate the autophagy activity. Data are expressed as the mean ± SD of seven biological replicates. Different lowercase letters indicate significant differences according to Duncan’s multiple range test (*p* < 0.05).

**Table 1 ijms-23-05464-t001:** Growth indexes of tomato seedlings 12 days after low nutrient stress treatment.

Treatment	Plant Height (cm)	Leaf Fresh Weight (g)	Root Fresh Weight (g)	Leaf Dry Weight (g)	Root Dry Weight (g)
WT/control	42.13 ± 0.31 b	48.75 ± 0.86 b	6.27 ± 0.43 b	3.78 ± 0.03 b	0.53 ± 0.02 b
OE/control	45.80 ± 0.46 a	54.34 ± 0.57 a	8.77 ± 0.43 a	4.57 ± 0.07 a	0.71 ± 0.03 a
WT/LNS	20.47 ± 0.25 d	9.15 ± 0.12 d	4.01 ± 0.21 c	1.58 ± 0.07 d	0.21 ± 0.03 d
OE/LNS	23.00 ± 0.26 c	11.18 ± 0.22 c	4.93 ± 0.35 bc	1.91 ± 0.03 c	0.39 ± 0.02 c

The data are shown as the means ± SDs of three independent biological replicates. Different lowercase letters in the same column indicate significant differences according to Duncan’s multiple range test (*p* < 0.05).

## Data Availability

The data that support the findings of this study are available from the corresponding author upon reasonable request. All relevant data can be found within the manuscript and its Appendix A.

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
