# Peer review of "Overexpression of PpSnRK1α in Tomato Increased Autophagy Activity under Low Nutrient Stress"

_ijms, 2022, doi:10.3390/ijms23105464_

Round 1

Reviewer 1 Report

Jingjing Luo and collaborators describe in this manuscript the effects of PpSnRK1α overexpression in tomatoes, improving the resistance of these plants to low nutrient stress. They show data that suggest this protective effect (which includes improved photosynthetic rates, better plant growth, and enhanced expression of antioxidant enzymes) is paired with the induction of autophagy by both inhibiting its major repressor (TOR) and mediating the ABA signaling pathway.

The study is generally well-performed, and the experiments are justified. However, some major points should be addressed before it could be considered for publication:

  • In my opinion, the authors should better explain the approach and context of this research, as well as its impact. For example, why overexpressing a foreign protein (Prunus persica) and not the one from tomato (Solanum lycopersicum) itself? Why is it convenient/useful/interesting to use this approach?

Also, readers would appreciate additional information in the introduction regarding SnRK1. What is the difference/analogy between SnRK1, SnRK2, SnRK1α, SnRK1.1…? What is KIN10? I believe the paper, and its importance, would be much more understandable if these issues are clarified.

  • Regarding the autophagy analysis, the results from the MDC staining seem clear, but more autophagy analyses are needed to clearly state that autophagy is induced. Even more considering that using MDC staining as a sole method to assess autophagy is not recommended (as discussed in Klionsky et al., Autophagy, 2021). Combining these data with additional experiments like the detection of autophagy markers by western blot or autophagic vesicles quantitation by electron microscopy is needed. Or, at least, include treatments with autophagy inhibitors to rule out the possibility that autophagosome accumulation is the result of autophagy blockage at a late stage, rather than its induction.

Also, increased expression of ATG genes may indicate autophagy induction, but it is not conclusive. Even more, if other ATG genes are downregulated.

Finally, it is not proved that autophagy induction is the cause of enhanced low nutrient tolerance. The authors should note this caveat or try overexpressing PpSnRK1α in an autophagy-deficient context (mutant plants, treatment with inhibitors) to assess the role of autophagy in this phenotype.

Minor points:

  1. On page 3, in the last paragraph, I believe “Fig. 2B” is wrong, as it should be referring to another figure.
  2. Figure 1B: I recommend labeling the X-axis as “Time of the day”.
  3. Figure 1C: Half of the legend is missing.
  4. Figure 2B: The legend is missing.
  5. Page 6, first line: Is it SlATG3 or SlATG3a, as stated in the figure?
  6. Figure 3B: Perhaps there is a problem with the version of the images that I have, but the green dots are overall weak. Also, I think pictures “OE control” and “WT LNS” are switched. Could this be the case?
  7. Page 11, last line of the first paragraph: I suggest changing the wording to “… enhanced the ABA-dependent activation of autophagy under LNS.”, as I believe it is easier to read this way.
  8. Figure 7A: I believe the colors of the legend do not match the colors of the data in the figure.
  9. I recommend the authors include additional information about the statistics either in the figure legends or in the “Materials and methods” section. Not only about the number of replicates, the sample sizes, or the tests they used (some of this information is already included in the figure legends), but also to better explain the use of lowercase letters to show significant differences. Sometimes, it is not clear which groups are compared, or what group is significantly different.
  10. There are some mistakes in the formatting of the first reference of the list.
  11. Table S1: The forward sequence of the primer for PpSlSnRK1α has a typo (a letter "F").

Reviewer 2 Report

Review: ‘Overexpression of PpSnRK1α in tomato enhanced low nutrient tolerance by regulating autophagy activity’.

In the present paper the Authors to analyze overexpression of PpSnRK1α in tomato. The manuscript is a solid study, very interesting and important for understanding of the connection between SnRK1 and autophagy.

On the growth of plant under stress, such as nutrient starvation resistance can be affected by the regulation of the SnRK1α. Further molecular and genetic approaches will accelerate our knowledge of PpSnRK1α functions, and inform the genetic improvement of nutrient starvation tolerance in tomato through genetic engineering.

The subjest of the manuscript fit the aims of the Journal and results could be interest for acientific community. However, in my opinion the Author should explain more detail:

  • to determination of SnRK1 activity
  • to measurement of photosynthesis rate and chlorophyll content
  • RT-qPCR total RNA extraction
  • extraction and ABA determination

Provide the literature of this position or a more detailed description of the procedures used, e.g. by means of HPLC

Round 2

Reviewer 1 Report

I appreciate all the corrections made by the authors, especially those that better explain the context and aim of this research.

I still believe that this study should not categorically state or conclude that autophagy induction results in higher tolerance, because a) more autophagy studies should be performed to unequivocally show autophagy induction and b) the phenotype should be also addressed in an autophagy-deficient context, to clearly show that autophagy is responsible for enhanced tolerance. It is very possible that this is ultimately the case, showing once again the beneficial effect of autophagy in cell and organism homeostasis, but conclusions like this one should be fully supported by more data. However, I totally understand this is a basic preliminary study, so at least these caveats should be included in the discussion before it is published.

Congratulations to the authors for this work and the paper, I very much enjoyed reading and reviewing it.

Author Response

Dear Reviewer,

Thank you for your comments concerning our manuscript entitled “Overexpression of PpSnRK1α in tomato enhanced low nutrient tolerance by regulating autophagy activity”.

We have carefully considered your comments and agree with you. Our research focused on the induction of autophagy by PpSnRK1α. The evidence that PpSnRK1α in tomato enhanced low nutrient tolerance by regulating autophagy activity is insufficient.

Therefore, we have revised the descriptions related to this in the “title”, “abstract”, “introduction”, “discussion”, and “summary” sections, hoping to get your approval.

Thanks again.

Kind regards.

Jingjing Luo